# Impact of Axial Eye Size on Retinal Microvasculature Density in the Macular Region

**DOI:** 10.3390/jcm9082539

**Published:** 2020-08-06

**Authors:** M. Hafi Khan, Andrew K. C. Lam, James A. Armitage, Lisa Hanna, Chi-ho To, Alex Gentle

**Affiliations:** 1School of Medicine, Faculty of Health, Deakin University, Waurn Ponds, VIC 3216, Australia; hafi.khan@hotmail.com (M.H.K.); j.armitage@deakin.edu.au (J.A.A.); 2Centre for Myopia Research, School of Optometry, The Hong Kong Polytechnic University, Kowloon, Hong Kong 00852, China; andrew.kc.lam@polyu.edu.hk (A.K.C.L.); chi-ho.to@polyu.edu.hk (C.-h.T.); 3School of Health and Social Development, Faculty of Health, Deakin University, Geelong, VIC 3220, Australia; lisa.hanna@deakin.edu.au

**Keywords:** refractive error, myopia, eye size, optical coherence tomography angiography (OCTA), retina, retinal perfusion, vascular perfusion

## Abstract

Mechanical tissue stresses are important contributors to the increased risk of sight-threatening pathology in larger, more myopic eyes. The contribution of altered ocular vasculature to the development of this pathology is less well defined. The current study investigated the impact of eye size on the superficial vasculature of the macula. Subjects (*n* = 104) aged 18–50, with no history of ocular or vascular disease, or myopia control, were recruited from university staff and student populations in Australia and Hong Kong. Refractive error, ocular size, retinal morphology and vascular morphology were quantified through open field autorefraction, ocular biometry and ocular coherence tomography angiography. Morphology of the superficial retinal capillary plexus was assessed over a 3 × 3 mm fovea-centred area. Perfusion area and vessel length densities were analysed relative to axial eye length and retinal thickness. A significant inverse association was found between axial length and vascular density measures (perfusion area density r^2^ = 0.186, *p* < 0.001; and vessel length density r^2^ = 0.102, *p* = 0.001). Perfusion area and vessel length densities were reduced by 5.8% (*p* = 0.001) in the longest, relative to the shortest, eyes. The aggregated ganglion cell layer inner plexiform layer thickness was also inversely associated with eye size (r^2^ = 0.083, *p* = 0.003), and reduced, by 8.1% (*p* < 0.001), in the longest eyes. An inverse association of eye size and superficial retinal vasculature density, that is not simply explained by retinal expansion or image magnification factors, was confirmed. These data support the hypothesis that ongoing metabolic challenges may underlie the development of myopia-related and -associated pathology in larger eyes.

## 1. Introduction

Globally, myopia was present in 28.3% of the population in 2010 and this is expected to rise to 52% by 2050 [1]. This rapid increase in global myopia prevalence, coupled with the increasing availability of clinical interventions and protocols to slow the progression of myopia [2], warrants a greater understanding of both the earliest signs of myopia development and the pathophysiologic consequences of the condition. Such an understanding will help clinicians better identify those that may benefit from myopia control interventions, and therefore combat the visual impairment associated with increasing degrees of myopia [1,3,4].

Excessive axial elongation is the primary mechanism for the development of myopia. The expansion of scleral surface area in the myopic eye results in the stretching and thinning of other layers of the globe, particularly the underlying choroid and retina, which probably leads to the pathological complications in these tissues [5]. Glaucoma, retinal detachment and myopic maculopathy are just some of the pathological conditions that are known to result [6,7].

The disease mechanisms of myopia-associated retinal conditions are still not fully understood and current knowledge of the relationship between myopia and retinal vascularity is very limited. This relationship is likely to be particularly important because refractive errors like myopia largely manifest between 8 and 14 years of age, whereas the pattern and density of the vascular supply to the retina is established at, or just after, birth [8]. Postnatal growth of the eye is not in proportion with other structures, especially the vascular density. It follows that, depending on how the retinal vascular supply responds to changing eye size, people who develop larger eye sizes may be served by a lower blood vessel density in their retina, due to the mismatch in vascular supply from birth and the significant eye size increase slightly later on in life [9], and that susceptibility to retinal disease may be higher as a result [10]. It is therefore important to better understand the link between eye size and retinal vascular density as the retinal vasculature must be a robust and efficient network to supply metabolic substrate to perhaps the most metabolically active tissue in the body. The per gram oxygen demand of retinal tissue has been described as being more than that of the brain, especially at the macular region [11]. It follows that optimal eye health and vision is highly dependent on an effective vascular supply and the retina is likely more susceptible to small deficits in vascular perfusion than many other tissues.

The primary aim of the current study was to investigate the relationship between axial eye size and the superficial retinal microvasculature, using a commercial optical coherence tomography angiography device, as defined in terms of vessel length density and perfused area density at the macular region. In addition, the study sought to identify any correlates or markers that indicated for either reduced retinal perfusion or impacts on the metabolic needs of the superficial retina in larger eyes, thus potentially demonstrating the clinical imperative for early childhood myopia control.

## 2. Methods

This study was conducted with approval from institutional human research ethics committees at Deakin University (HEAG-H 33_2018) and Hong Kong Polytechnic University, (HSEARS20180405001), the sites at which subjects were recruited. All subjects gave informed consent prior to participation in the study and did not receive financial compensation. This study adhered to the tenets of the Declaration of Helsinki.

### 2.1. Study Participants and Recruitment

In total, 104 healthy individuals aged 18–50 years were included, in order to ensure that the retinal microvasculature was assessed during a period of adulthood where axial eye length changes might be expected to be absent or minimal [12]. Subjects with major ocular and retinal pathology (such as, but not limited to, significant retinal lesions, significant cataract, glaucoma and retinal detachment), the presence of vascular disease (conditions such as diabetes and ocular vascular occlusions), or a history of premature birth (prior to 36 weeks of gestation) were excluded, as were those with a history of myopia control. Those with myopic degeneration were not excluded but the presence of such was noted (19 myopic subjects displayed myopic peripapillary changes, but no other myopic degenerative signs were found in the study population).

### 2.2. Hypothesis and Sampling

The findings of the current study were considered in terms of a very simplistic conceptual model that made assumptions about how a vascular network pattern embedded in the surface of a sphere might alter as the sphere expanded, effectively simulating eye growth. It was reasoned that, in the presence of vascular remodelling to accommodate vessel extension, and the absence of either compensatory angiogenesis or vascular loss as eye size increased, the vessel length density parameter should remain very approximately unchanged as vessels elongated in concert with the expansion of the surface on which they lay. It was also reasoned that, depending on the capacity of the vessels to maintain, or increase, their calibre in an enlarging eye, the perfusion area should also remain approximately constant in larger eyes.

All participants were categorised into one of five groups based on their axial eye length, with eye length groupings defined using the methods of Bueno-Gimeno, Espana-Gregori [13], Foster, Broadway [14], Lim, Cheung [15], Mutti and Hayes [16]. The axial eye size bins used were defined and named as follows: ‘hyperopes’ < 23.50 mm; ‘emmetropes’ ≥ 23.50 mm and ≤24.50 mm; ‘low myopes’ > 24.50 and ≤25.50 mm; ‘moderate myopes’ > 25.50 mm and ≤26.49 mm; and ‘high myopes’ ≥ 26.50 mm. This approach was justified in terms of the study expectation, that mechanical effects of axial eye size, not the refractive error per se, impact retinal microvasculature density, either passively, actively or both, in the macular region of the human eye. However, as seen later in the study (Table 1) the eye size bins do not correlate perfectly with the refractive naming convention applied to each bin, due to the natural variations in the size of eyes with a given refractive error and the effect of cylindrical refraction [17]. This being said the correlation between axial eye length and refractive error for participants in this study (*r* = 0.859, *p* < 0.001 for axial length and *r* = 0.847, *p* < 0.001 for Vitreous Chamber Depth (VCD)) were in line by those reported for previous studies [18].

### 2.3. Ocular Health, OCTA Imaging, Refraction and Biometry

A questionnaire, consistent with the inclusion/exclusion criteria of the study, was used to collect history of ocular and vascular health. Retinal photography (fundus camera, Visucam 200, Carl Zeiss, Jena, Germany) was performed to image the central 40 degrees of the retina and allow detection of any undiagnosed retinal complications. This was followed by open-field auto-refraction (NVision K-5001, Shin-Nippon, Kagawa, Japan), ocular biometry (LenStar LS 900 Ocular biometer, Haag-Strait Diagnostics, Koeniz, Switzerland), and Optical Coherence Tomography Angiography (OCTA; Cirrus 5000 with AngioPlex, Carl Zeiss, Jena, Germany). Subjects with best corrected visual acuity worse than logMAR 0.10 (6/7.5; 20/25) were also excluded before final analysis. All data acquisition was performed by the same operator. The equipment used was replicated at both sites.

For objective refraction, the average of five readings per eye was used to determine the mean spherical equivalent (MSE). Results within ±1.00 D for the spherical and ±0.50 D for the cylindrical component were accepted or otherwise repeated. The average of five biometry measurements was used to determine axial length and VCD. OCTA was used to image five successive angiography scans, each over a 3 × 3 mm macular area centred on the fovea. OCTA scans of signal strength ≥7 were accepted and parameters were derived from the average of five scans [19,20]. Live-tracking with the instrument’s built-in software allowed motion artefact-free data to be generated. Although all data were captured for both eyes, only data for the longer eye of each subject were used in the final analysis.

### 2.4. Statistical Analysis

Linear regressions, analysis of variance (ANOVA) and Kruskal–Wallis tests were performed to determine the relationship between ocular axial parameters and retinal vasculature density. Two separate measures of retinal vessel density were outputted by the Cirrus 5000: vessel length density, being the total length of perfused blood vessels per unit area in the region of measurement and a surrogate for change in capillary length as vessel width does not influence the measurement; and perfusion area density, being the total area of perfused vasculature per unit area in the region of measurement and a surrogate for overall capillary size as both vessel length and vessel calibre influence the measurement. The linear regressions determined the effect of axial length on superficial retinal vasculature across an aggregated foveal and para-foveal vascular area. This measured area was centred at the fovea and consisted of a 3mm radial annulus. One-way ANOVA was performed for normally distributed data to assess differences in vascular parameters between the groups listed in Table 1. The macula was then divided into sub-regions to test for regional associations (Figure 1). For non-parametric data, the Kruskal–Wallis test was used. Bonferroni adjustments were applied where multiple comparisons were made. The Chi-square test was applied to analyse gender and laterality effects across groups. The Kolmogorov–Simonov test was used to test for normality.

## 3. Results

A total of 104 healthy subjects (104 eyes) were included in the analysis. They were divided in 5 subgroups based on axial length measurements outlined previously. The number of subjects in each group along with their demographic characteristics and outcome measures are summarised in Table 1.

All overall experimental outcomes suggested a decrease in vascular density at the macular region (perfusion area and vessel length densities) as axial eye size parameters increased (Figure 2, Figure 3, Figure 4 and Figure 5). Perfusion area density was defined as the percentage of total area of perfused vasculature per unit area in a region of measurement, whereas vessel length density was defined in mm^−1^ as the total length of perfused vasculature per unit area in the region of measurement. Figure 2 and Figure 3 show that as the eye size increased (axial length), the area of perfused vasculature at the macular region decreased significantly (*p* < 0.001). The results showed that the aggregated length of those vessels also decreased as the eye size increased (*p* < 0.001). Additional analysis of the data showed that this change was not also related to either age (*p* = 0.34 (perfusion area density); *p* = 0.18 (vessel length density)) or gender (*p* = 0.15 respectively).

As axial eye enlargement in myopia mostly occurs in the vitreous chamber, and the other main parameters of the axial length (anterior chamber depth and lens thickness) contribute relatively little to the eye size changes, the association of the vascular measures with VCD was also tested [21]. Figure 4 and Figure 5 show the association between VCD and both of the vascular measures. As the VCD increased, there was a statistically significant decrease (*p* < 0.001) in perfusion area and vessel length densities at the macular region, as was the case for axial length.

Overall, there was a significant reduction is macular perfusion area density and vessel length density in the longest eyes (‘high myopes’), compared with the shorter eye groups (specifically the ‘emmetropes’). The decreases were −2.24% (*p* < 0.001) and −1.25 mm^−1^ (*p* < 0.001), respectively. In absolute terms, this difference equates to a 5.8% decrease in perfusion area density and a 5.8% difference in vessel length area density in the longest eyes compared with the emmetropic eyes (Table 2).

While the overall analysis of aggregated and para-foveal regions is important to determine the vasculature relationships with the different axial lengths, this may mask subtle local differences in vascular density distributions. Given the highly significant association between axial length and para-foveal vascular density, this region was further divided into nasal, temporal, superior and inferior divisions for analysis. Significant reductions in both perfusion area density and vessel length density were found in all four para-foveal quadrants when high myopes were compared with emmetropes (Figure 6). However, no significant differences, in the amount of reduction in vascular density, were identified between quadrants (*p* > 0.05).

Foveal avascular zone (FAZ) area and boundary circularity were also analysed. Interestingly, and despite the significant vascular perfusion density and length differences with eye size, there were no significant associations between axial length and FAZ area (r^2^ = 0.004, *p* = 0.545) or FAZ boundary (r^2^ = 0.0001, *p* = 0.846). Notably, however, even though overall macular thickness showed no association with superficial retinal vascular density (r^2^ = 0.011, *p* = 0.291), a weak, but significant, relationship was found between the Ganglion Cell Layer-Inner Plexiform Layer (GCL-IPL) complex and superficial vascular density (Figure 7, r^2^ = 0.083, *p* = 0.003).

## 4. Discussion

The current study sought to determine the relationship between the density of the superficial retinal microvasculature, in the macular region of the human eye, and postnatal axial eye size. In doing so, the study also sought to identify data patterns that indicated whether the retinal vascular tree enlarges in concert with retinal expansion in larger eyes, or showed signs of active angiogenesis or vascular loss either to compensate for the larger eye size or as a consequence of it. The study data suggested that longer eyes showed both a decreased superficial perfusion area and a decreased vessel length density. In addition, although overall macular thickness did not change significantly, a significant decrease in the GCL-IPL complex was found in larger eyes. When compared against the simple model that was devised to consider study outcomes, the reduced vessel length density data may suggest that relative vessel loss and reduced retinal perfusion is a feature of larger eyes and over time, and in the largest eyes, presents metabolic challenges that impact retinal structure and function. However, the data may also be interpreted as indicating that compensatory vessel growth has simply not occurred across the increased surface area of the retina in the enlarged eye, albeit to the same metabolic endpoint. Such observations serve to emphasise the importance of offering early clinical intervention aimed at preventing or controlling the development of any myopia.

In this study, we used OCTA technology to quantify vessel density of the superficial retina and examine its relationship with axial elongation. It is presumed that the endowment of retinal vascular supply is determined by a developmental assumption of final eye size, and that the majority of angiogenesis is completed in the first years of life [22]. However, in the majority of individuals who ultimately become myopes, the eye starts to elongate excessively at a time (usually age 8–14 years) post-angiogenesis and therefore these longer eyes may develop a reduced retinal blood vessel density, thus potentially exposing the retina to metabolic challenge [22]. This study demonstrated an inverse association between axial length and vasculature density within the superficial macular vascular plexus (Figure 1), at the very least demonstrating the potential for vascular compromise in larger eyes. This finding was derived from data across the refractive continuum, and in a large, heterogeneous population. Post-hoc analysis demonstrated significantly decreased perfusion and vessel density in high myopes relative to emmetropes of approximately 5% (Table 2), which for a tissue as metabolically active as the retina must be considered a significant relative deficit. Significantly, there was found to be no observable relationship between overall macular thickness and axial length; however, as shown in Table 1, the GCL-IPL thickness significantly decreased as axial length increased. This suggests that changes in the retinal layers were local to the vascular network investigated.

Given the relatively modest nature of the numerical change measured, it is important to consider whether the methodology can reliably detect such a change, and a recent publication from our laboratories demonstrates the capability of the repeated measurement paradigm employed in the current study to detect this relatively small difference [20]. In addition, differential magnification factors in smaller and larger eyes are likely to have an impact on the measured vascular densities. Sampson and Gong [10] reported that adjustment algorithms may be necessary to normalise measured densities for comparison across different eye sizes. In the current study, the instrument used employed an axial length of 23.80 mm as the reference; however no correction was made to measurements based on axial length as: (1) the 3 × 3 mm field employed is small enough to be relatively robust to magnification variations with eye size (maximum of 5% variation); and (2) the image magnification effect in larger eyes results in an increase in vascular density; thus, the deficits we report here are, if anything, an underestimation of the true deficits in larger eyes [10].

A small number of studies to date have examined the relationship between retinal vascular morphology and ocular axial length and/or refractive error. Wang and Kong [23] were unable to demonstrate any significant relationship between retinal vessel density and axial length, also suggesting that their findings were previously also demonstrated by Benavente-Perez and Hosking [24]. However, it should be noted that the latter authors used retinal flowmetry to measure blood flow and volume, rather than OCTA. Despite the lack of a defined relationship, Wang and Kong [23] did find significantly lower vessel density in high myopes in the peri-papillary region, which is more broadly consistent with the findings of the current study. Additionally, Wang and Kong [23] investigated a younger cohort (approximate average age 16.5 years) in which final eye size may not have been fully established [7]. Shimada and Ohno-Matsui [25] found a reduction in retinal blood flow in longer eyes compared with normal eyes but used older technology than the currently available OCTAs. Al-Sheikh and Phasukkijwatana [26] and Fan and Chen [27], however, demonstrated reduced vascular density associated with increase in axial length. Leng and Tam [28] recently also demonstrated reduced superficial retinal vessel density in longer eyes, but used both eyes of their subjects in their statistical analysis and investigated a broader age range that included both much younger and much older subjects than the current study (range 8 to 87 years). Suwan and Fard [29] studied the microvasculature at the peri-papillary region, instead of the macula, and found no significant change in capillary density between their control and myopia groups. However, they did demonstrate a significant reduction in perfused capillary density in myopic eyes that were also glaucomatous, when compared with the control population. So, to summarise, previous studies have been equivocal in establishing a link between axial length and vascular perfusion of the superficial retinal layers, although findings that are broadly supportive of those in the current study have been reported. However, the studies referred to above used relatively small sample sizes, compared with the current study, and some of those studies used external image analysis over machine derived quantitative metrics.

The mechanism and consequences of reduced vascular density in longer eyes has yet to be established. Based on current observations and known ocular physiology, there are three plausible models suggested by the phenomena observed in studies to date [23,26,29,30,31]:

Model 1: Increasing eye size causes stretching and redistribution of the retina, in turn increasing the total retinal area. The total number of retinal ganglion cells present in the retina, and the existing retinal vessels, therefore spread over a larger area which decreases the density of ganglion cells, or the count per unit area. This model holds that it is entirely physiological for the eye to exhibit reduced vessel density as these vessels are supplying a reduced density of ganglion cells. In this model, the retina can be assumed safe from any metabolic consequence of reduced vascular density. Al-Sheikh and Phasukkijwatana [26] and Milani and Montesano [32] have both presented results that support this model, finding significantly reduced macular vessel density in their myopic groups and suggested that straightening and narrowing of blood vessels due to excessive eye elongation results in overall reduced vessel density.

Model 2: Larger eyes have fewer ganglion cells overall and therefore fewer vessels are needed to provide metabolic substrate, blood and nutrients. A supply–demand principle therefore explains the reduced vascular density and implies an active change. The prediction for these eyes is that physiological mechanisms are at play rather than pathological. Fan and Chen [27] demonstrated reduced superficial macular vascular density associated with increased axial length. Fan and Chen [27] suggested that because their highly myopic group had reduced ganglion cell complex (GCC: nerve fibre layer, GCL and IPL complex) thickness, retinal layers that are supplied by the superficial vessels, the lower number of ganglion cells in longer eyes reduces the oxygen demand and therefore vessel density in this region. This idea of supply-demand was used by Mo and Duan [31] to explain the finding that there was no significant difference between emmetropes and high myopes in the superficial or deep macular region. Mo and Duan [31] speculated excessive elongation causes thinning of the retinal tissue, which might decrease the oxygen demand and consequently decrease the blood perfusion and vessel density.

Model 3: Stretching and thinning of the retina in myopia ultimately results in some vascular compromise and reduced vascular density which, in turn, leads to a net loss of ganglion cells. These individuals have previously had a greater number of retinal ganglion cells but the lack of vascular supply resulted in cell death and tissue remodelling. This model predicts the progressive development of a physiological challenge in larger eyes that, ultimately, may lead to metabolic failure.

We are inclined to suggest that the results of the current study are best explained by this third model, in that that both vascular perfusion density and vessel length density were as much as 5% less in larger eyes, and likely more were magnification adjustment factors to also be applied [10]. In addition, a reduction was also found in the GCL-IPL complex in larger eyes, despite relatively little change in overall retinal thickness in the area investigated, which may be interpreted as localised tissue loss inconsistent with findings in the other, associated retinal layers. Indeed, findings in high myopes suggest that contrast sensitivity and spatial summation deficits in larger eyes may be a result of post-receptoral remodelling in larger eyes [33]. Given that both of these psychophysical properties are mediated by ganglion cells of the macular region [34,35], this supports the argument that reduced superficial vascular density may lead to remodelling of the GCL-IPL complex and altered retinal function.

Ganglion cell loss and reduced GCC thickness are also characteristics of glaucoma development [36]. It may also, therefore, be reasonable to suggest reduced macular perfusion in larger eyes plays a role in glaucoma development as ganglion cell count decreases because of a metabolic challenge. While the present study aimed to discover the association of myopia with vasculature and not the vascular consequences manifesting as glaucoma, this is important to consider in the broader context of clinical implication of the current findings. The pathophysiology of glaucoma remains not fully understood and, given the risk factor links between myopia and glaucoma current findings loosely link a vascular insufficiency in myopia with the potential for development of conditions such as glaucoma.

The work by Suwan and Fard [29] does echo some of the explanations we have suggested above. They suggest that (1) myopic elongation stretches the retinal tissue, resulting in decreased vascular endothelial growth factor production and a putative decrease in retinal microvascular density [29,37]; (2) the decrease is vascular density may be a result of reduced oxygen demand because of myopic retinal degeneration; (3) RNFL loss may reduce regional oxygen demand, thus triggering retinal vascular attenuation via auto-regulatory mechanisms [23,29]. However, it must be acknowledged that Suwan and Fard [29] based their discussion on results acquired from the peripapillary, not macular, region.

The superficial vascular plexus quantified in the current study supports displaced amacrine cells, ganglion cell bodies and fibres. Metabolic dysfunction within the vascular networks at the superficial plexus may put ganglion cell fibres under metabolic stress, making them more vulnerable to the environmental challenges of intra-ocular pressure (IOP) excursions or perfusion deficits that may underpin the development of glaucoma. Moderate myopia is a significant risk factor for glaucoma and it also preferentially affects the superficial retina in the macular more than the deeper retinal layers [22,38]. Although glaucoma is considered a multifactorial condition, there remains a case for the presence of a vascular component in the development of glaucoma. We have demonstrated that a reduced macular perfusion area and associated GCL-IPL thickness reduction, which might indicate that ganglion cell loss has occurred, may be found larger eyes, but it remains to be demonstrated whether this phenomenon is in part explanatory of the link between myopia and glaucoma risk.

Previous studies identified age-related changes in retinal vascular density, with a reduction in vascular density reported in older populations across the two studies [39,40]. We found no significant changes in vascular density to be associated with either age or gender; however, the specific parameters of our experimental design precluded the assessment of an equivalent age-range to these studies, which may explain why any effect of age did not reach significance in the current study.

We have identified two main limitations in the methodology of our study. Firstly, we have not corrected our data for potential magnification artefacts in different eye sizes [41], although we have assumed that, in line with the observations of Sampson and Gong [10], our findings are likely an underestimate of the effects of eye size on retinal vascular density. Secondly, our conclusions are limited by the fact that our OCTA instrument system only allowed us to measure the effect of eye size on the superficial retinal vascular plexus. We speculate that deeper retinal plexuses would be subject to similar, or even more significant, expansional effects due to the larger surface area of the deeper ‘retinal shells’ and demonstration of a similar phenomenon in the deeper vascular plexus would imply the metabolic insult to the retina is more widespread than implied by the current findings.

## 5. Conclusions

The key outcome of this study is the demonstration of the reduced retinal perfusion area and vessel length density in larger eyes compared with shorter eyes. The inverse association of axial length with vascular density is contrary to the prior expectations of the simple retinal and vascular stretch in enlarged eyes model employed, and has not been demonstrated previously. The possibility that reduced perfusion density in high myopia is merely physiological, as it is a consequence of retinal stretching and thinning, is a less than satisfactory explanation of current findings. There is a stronger case that reduced vascular density leads to metabolic stresses in the retina that may predispose to the sight-threatening pathological complications seen in higher degrees of myopia. This observation strongly supports the importance of offering clinical myopia control interventions either prior to, or early in, the process of childhood myopia development.

## Figures and Tables

**Figure 1 jcm-09-02539-f001:**
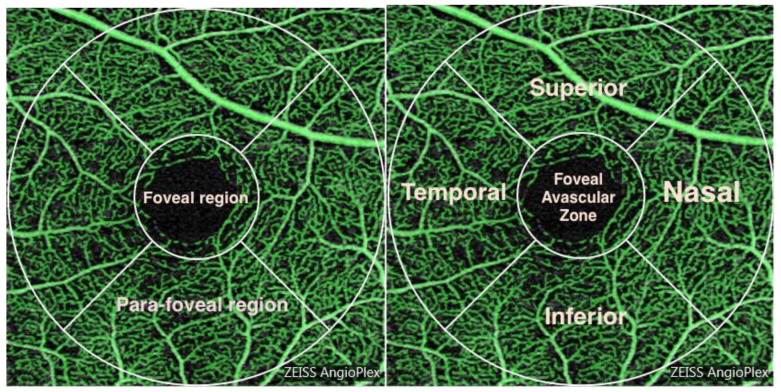
Pictorial representation of the macular vasculature. The vessels in a 3 mm radius annulus centred at the fovea were measured. Division of the annulus into foveal and para-foveal regions is shown along with further division into superior, inferior, nasal and temporal quadrants. The radius of the aggregated foveal/parafoveal region is 3 mm whilst the foveal region has a 1mm radius.

**Figure 2 jcm-09-02539-f002:**
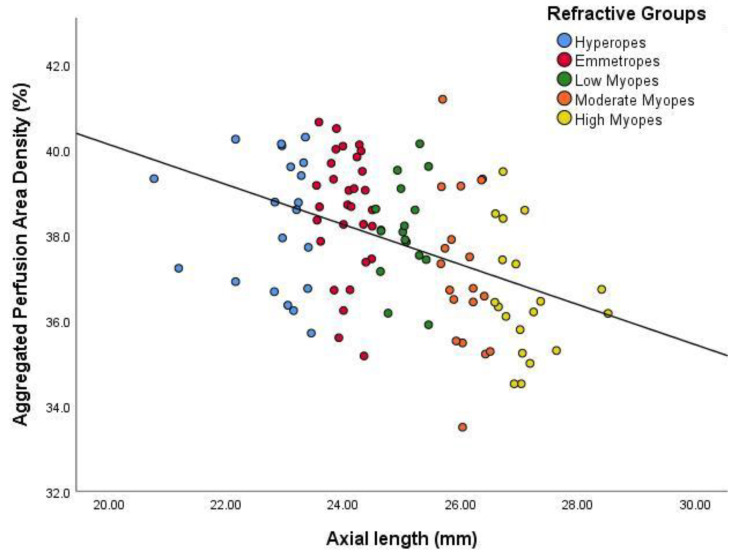
Scatter plot demonstrating the relationship between aggregated perfusion area density and axial length. The colours distinguish the subjects for their particular eye size bin. The regression line indicates a significant inverse relationship (r^2^ = 0.186, *p* < 0.001).

**Figure 3 jcm-09-02539-f003:**
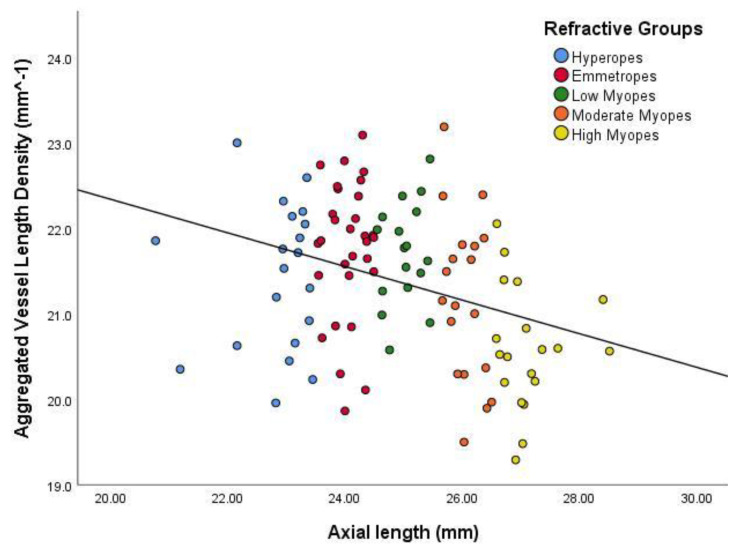
Scatter plot demonstrating the relationship between aggregated vessel length density and axial length. The colours distinguish the subjects for their particular eye size bin. The regression line indicates a significant inverse relationship (r^2^ = 0.102, *p* < 0.001).

**Figure 4 jcm-09-02539-f004:**
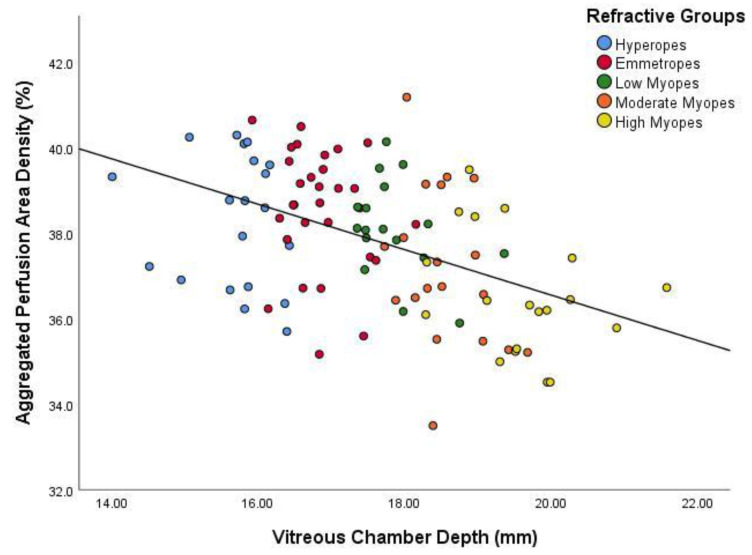
Scatter plot demonstrating the relationship between aggregated perfusion area density and vitreous chamber depth. The colours distinguish the subjects for their particular eye size bin. The regression line indicates a significant inverse relationship (r^2^ = 0.225, *p* < 0.001).

**Figure 5 jcm-09-02539-f005:**
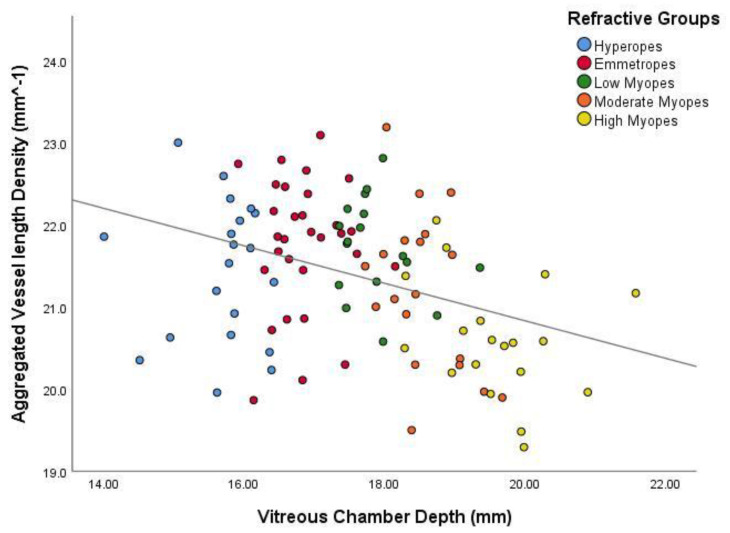
Scatter plot demonstrating the relationship between aggregated vessel length density and vitreous chamber depth. The colours distinguish the subjects for their particular eye size bin. The regression line indicates a significant inverse relationship (r^2^ = 0.134, *p* < 0.001).

**Figure 6 jcm-09-02539-f006:**
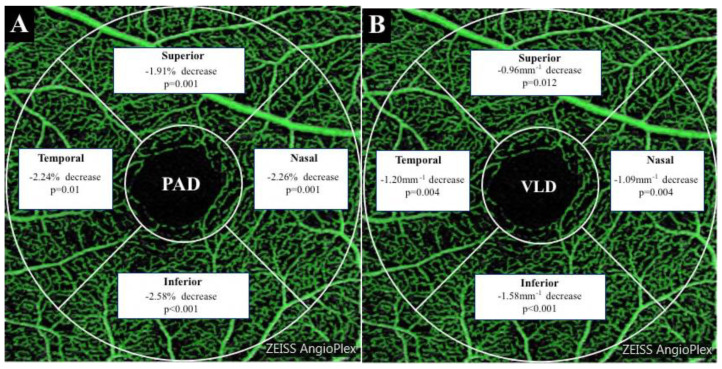
Summary of regional variations in vascular Perfusion Area Density (PAD) (**A**) and Vessel Length Density (VLD) (**B**) between emmetropes and high myopes. These are based on para-foveal region divided in four sectors as per the Early Treatment Diabetic Retinopathy Study grid.

**Figure 7 jcm-09-02539-f007:**
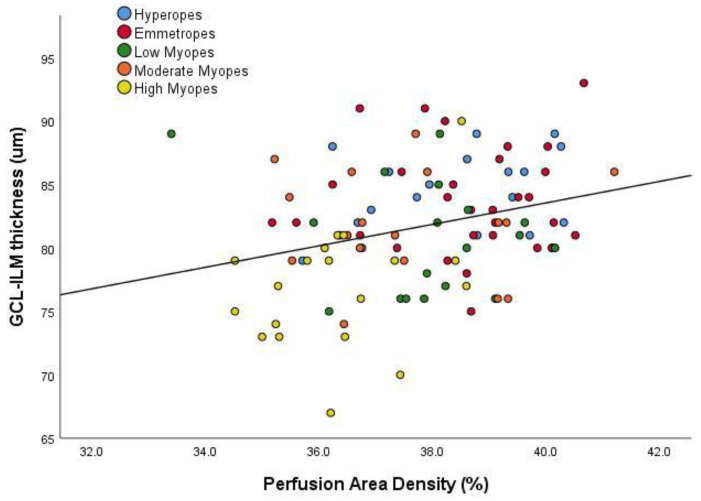
Scatter plot demonstrating the relationship between GCL-IPL Ganglion Cell Layer-Inner Plexiform Layer. The colours distinguish the subjects for their particular eye size bin. The regression line indicates a significant relationship (r^2^ = 0.083, *p* = 0.003).

**Table 1 jcm-09-02539-t001:** Demographic characteristics and outcome measures of participants between the refractive groups.

Descriptive(s)	‘Hyperopes’ (<23.50 mm)	‘Emmetropes’ (≥23.50 & ≤24.50 mm)	‘Low Myopes’ (>24.50 & ≤25.50 mm)	‘Moderate Myopes’ (>25.50mm & ≤26.49 mm)	‘High Myopes’ (≥26.50 mm)	*p* Value
N (=104)	20	30	18	17	19	
Age (years)	21 (5)	21 (4)	21 (5)	21 (10)	21 (21)	0.940
MSE (D)	**+0.07 (1.59)**	−1.44 (2.49)	**−3.40 (4.14)**	**−5.00 (2.38)**	**−7.12 (3.63)**	<0.001
Axial Length (mm)	**22.84 ± 0.73**	24.05 ± 0.30	**25.0 ± 0.32**	**26.02 ± 0.27**	**27.10 ± 0.55**	<0.001
Vitreous Chamber Depth (mm)	**15.69 ± 0.63**	16.85 ± 0.49	**17.88 ± 0.52**	**18.44 ± 0.51**	**19.63 ± 0.81**	<0.001
Gender (Male:Female = 40:64)	4:16	11:19	7:11	8:9	10:9	0.281
Laterality (OD:OS = 69:35)	16:4	21:9	11:7	10:7	11:8	0.540
Macular thickness (Aggregated; µm)	309 ± 13.0	309 ± 13.7	307 ± 16.2	312 ± 13.5	308 ± 14.0	0.490
Macular thickness (Fovea; µm)	256 ± 23.5	253 ± 19.5	249 ± 22.6	256 ± 17.4	254 ± 19.6	0.822
Macular thickness (para-fovea; µm)	315 ± 12.9	316 ± 13.6	315 ± 15.9	317 ± 14.0	305 ± 14.7	0.859
Perfusion Area (aggregated; %)	38.3% ± 1.53	38.6% ± 1.42	37.9% ± 1.56	37.5% ± 1.64	**36.3% ± 1.26**	<0.001
Perfusion Area (fovea; %)	21.9 ± 5.95	22.7 ± 4.53	20.6 ± 4.69	21.2 ± 4.16	20.4 ± 4.13	0.439
Perfusion Area (para-fovea; %)	40.4% ± 1.48	40.6% ± 1.34	40.1% ± 1.48	39.6% ± 1.59	**38.3% ± 1.2**	<0.001
VLD (aggregated;—mm^−1^)	21.4 ± 0.86	21.7 ± 0.81	21.6 ± 0.92	21.4 ± 0.87	**20.5 ± 0.68**	<0.001
VLD (fovea;—mm^−1^)	12.5 ± 3.50	13.1 ± 2.56	12.1 ± 2.79	12.6 ± 2.30	12.2 ± 2.35	0.727
VLD (para-fovea;—mm^−1^)	22.6 ± 0.81	22.9 ± 0.76	22.8 ± 0.88	22.5 ± 0.87	**21.7 ± 0.76**	<0.001
GCL-IPL thickness (µm)	84 (6)	83 (5)	80 (7)	81 (9)	**77 (6)**	<0.001

Numbers appear as Mean ± standard deviation for normally distributed variables and as Median (IQR) for non-parametric data. MSE indicates Mean Spherical Equivalent, VLD indicates Vessel Length Density and GCL-IPL indicates Ganglion Cell Layer-Inner Plexiform Layer. All statistically significant differences (*p* < 0.05), relative to the ‘Emmetropes’ group are indicated in bold. Normally distributed data were analysed by one-way ANOVA and non-normal data analysis by Kruskal-Wallis. Chi-square test was applied to analyse the frequency of data for non-continuous descriptive(s) such as gender and laterality.

**Table 2 jcm-09-02539-t002:** Multiple comparisons (post-hoc) of vascular density between short versus long eyes.

Outcome Measures	‘Emmetropes’ vs ‘High Myopes’	
	Mean absolute difference (*p*-value)	Relative difference (%)
Perfusion Area (aggregated; %)	−2.24 **(*p* < 0.001)**	5.8%
Perfusion Area (para-fovea; %)	−2.24 **(*p* < 0.001)**	5.5%
VLD (aggregated;—mm^−1^)	−1.25 **(*p* < 0.001)**	5.8%
VLD (para-fovea;—mm^−1^)	−1.20 **(*p* < 0.001)**	5.2%
GCL-IPL thickness (µm)	−6.79 **(*p* < 0.001)**	8.1%

The mean difference is significant at the 0.05 level and –ve denotes decrease, significant *p*-values are bolded. The differences between groups were tested with the Bonferroni post-hoc adjustments. VLD = Vessel Length Density.

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
