# Peer review of "Impact of Axial Eye Size on Retinal Microvasculature Density in the Macular Region"

_jcm, 2020, doi:10.3390/jcm9082539_

Round 1

Reviewer 1 Report

In this manuscript, the authors describe the results of their study examining the effect of axial eye length (myopia) on superficial retinal vasculature.  They show that as the degree of myopia increases, vascular length and vascular perfusion area decreases, as does the size of the GCL-IPL region.  The authors conclude that the change to the retina results from changes in vascular density, which compromises retinal cells.  The results suggest clinical treatment of myopia should occur early to prevent worsening complications.  Overall, the paper is well-written and the results are clear.

Major comments:

  1. Table 1 – the authors should highlight the significant p-values, in addition to where the significant differences are located.
  2. Participants in the study cohort covered quite large range of ages (8 - 80) – were any age-related differences noted? If so, what where they?
  3. Related to the above point about participant ages, according to Table 1, there is a discrepancy in study numbers. The N is listed as 104, but if you look at the ages listed, all participants were all in their early 20’s with an N = 45. Please clarify this difference.
  4. What is the difference between perfusion area and vessel length area? These seem to be the same things and results for both are very similar (which makes sense since one is directly related to the other). Perfusion length area may be the better length since it speaks to the function of the vessels and tissue health.
  5. In the first part of the discussion, line 73, data ‘suggest that relative vessel loss’. Was there vessel loss? Or was it the case that no new vessel formed?
  6. The authors mention that they only measure superficial vasculature. Can the authors speak to what they think is happening in the inter-retinal plexi, given that he tissue is stretching and they see changes in inner retinal thickness.
  7. The authors indicate that Model 3 (discussion) is occurring, which is a net loss of ganglion cells. Since the authors were examining the macular area, did they notice any changes to the macula specifically (or the optic nerve) that would indicate GC loss?

Author Response

Reviewer 1

Comments and Suggestions for Authors

In this manuscript, the authors describe the results of their study examining the effect of axial eye length (myopia) on superficial retinal vasculature.  They show that as the degree of myopia increases, vascular length and vascular perfusion area decreases, as does the size of the GCL-IPL region.  The authors conclude that the change to the retina results from changes in vascular density, which compromises retinal cells.  The results suggest clinical treatment of myopia should occur early to prevent worsening complications.  Overall, the paper is well-written and the results are clear.

Major comments:

Table 1 – the authors should highlight the significant p-values, in addition to where the significant differences are located.

We thank the reviewer for the suggestion. As a result, we have bolded all significant values, relative the ‘Emmetropes’ group, in Table 1 (Page 5 of 15) and modified the associated figure legend accordingly to reflect this. We have not provided all significant values in the table as we felt that the complexity of including all post hoc between group comparisons in the table may make the table particularly confusing for the reader.

Participants in the study cohort covered quite large range of ages (8 - 80) – were any age-related differences noted? If so, what where they?

The age range of the participants in our study was actually 18-50 years (Page 2 of 15, Line 73), however, as suggested by the reviewer, we have analysed our data set to determine whether either perfusion area density or vessel length density were associated with age. No significant associations were found between subject age and either of the vascular parameters, and we have indicated this in the results section of the manuscript (Page 6 of 15, Lines 8-10. Although there have been previous reports of associations between vessel density and age, the chosen experimental design precluded our cohort from including significant numbers of individuals from age ranges in which this change might have been expected (see more below under response to Reviewer 3).

Related to the above point about participant ages, according to Table 1, there is a discrepancy in study numbers. The N is listed as 104, but if you look at the ages listed, all participants were all in their early 20’s with an N = 45. Please clarify this difference.

The reviewer appears to have interpreted the median age and interquartile range of each group of participants, in row 2 of Table 1 as reflecting, the number of participants. However the numbers of participants in each group is indicated in the first row of the table and these do indeed total 104. We trust that the statement highlighted in the figure legend that indicates that non-parametric data is specified in the table as Median (IQR) is sufficient to clarify this point?

What is the difference between perfusion area and vessel length area? These seem to be the same things and results for both are very similar (which makes sense since one is directly related to the other). Perfusion length area may be the better length since it speaks to the function of the vessels and tissue health.

We thank the reviewer for pointing out the lack of clarity in our description of the two vessel density metrics in the study and have modified the description accordingly (Page 3 of 15, Lines 129-132). We also agree with the reviewer’s observation, that vessel length density may represent a better indicator of the health of the vascular tree, as it speaks directly to the changes in the vascular tree independent of any local vasoconstriction or vasodilation. However we would point out that, as the reviewer comments, the findings are very similar regardless of which parameter is analysed, which we argue likely speaks to the robust nature of the findings.

In the first part of the discussion, line 73, data ‘suggest that relative vessel loss’. Was there vessel loss? Or was it the case that no new vessel formed?

We agree with the reviewer, that either interpretation is viable (i.e. that either relative vessel loss or a lack of compensatory vascular growth may explain the experimental findings) and that the cross-sectional nature of the study does not allow us to make this distinction. We have therefore modified the discussion accordingly (Page 9 of 15, Lines 77-79).

The authors mention that they only measure superficial vasculature. Can the authors speak to what they think is happening in the inter-retinal plexi, given that he tissue is stretching and they see changes in inner retinal thickness.

Whilst our study is limited by our capacity to only quantify the superficial retinal vascular plexus, other instruments do facilitate such quantification and it would have been interesting to undertake such measurements. However, it is also the case that measurements of the deeper vascular plexi are prone to error through projection artefacts. Despite this, we speculate that the effects of the larger eye may be even more pronounced on the deeper plexi (due to the increased surface area of progressively deeper ‘retinal shells’) and have added a brief comment to this effect (Page 12 of 15, Lines 216-218)

The authors indicate that Model 3 (discussion) is occurring, which is a net loss of ganglion cells. Since the authors were examining the macular area, did they notice any changes to the macula specifically (or the optic nerve) that would indicate GC loss?

The reviewer raises an interesting point and we have indeed reported the significant change found in the GCL-IPL thickness in Table 1 (Pg 5 of 15, final row of table) and mentioned this where highlighted in the discussion (Pg 9 of 15, Ln 73; Page 12 of 15, Ln 202). We had elected not to make too much of this finding in the context of ganglion cell loss, as we cannot confirm that this thickness change actually reflects this. However we have added a brief comment indicating that this finding may indicate for ganglion cell loss in larger eyes (Pg 12 of 14, Line 203).

Reviewer 2 Report

The study aims at the relationship between axial eye size and microvasulature and here are some of major concerns;

  1. Sample size is too low (only 104) and out of 104 only 36 are moderate and high myopes’
  2. The authors excluded subjects with retinal pathology but it is also mentioned that myopic degeneration was not excluded in the sampling process. This makes a deviation from the sampling criteria and the authors didn’t mention in which group these participants come under.
  3. Age group is not convincing, the authors already mentioned eye elongation extensively occurs during the age of 8-14, then presence of participants up to 50 years again makes a confusion.
  4. In the page 3, Line number 120; it is mentioned that only the data for longer eye of each subject was used; that is not a proper way of analysis

Author Response

Reviewer 2

Comments and Suggestions for Authors

The study aims at the relationship between axial eye size and microvasulature and here are some of major concerns;

Sample size is too low (only 104) and out of 104 only 36 are moderate and high myopes’

We originally designed the study assuming that we would find a 15% difference in retinal vessel density between an emmetropic and a highly myopic eye, on the basis of the theoretical change in retinal area between the two eye sizes. We calculated that a sample size of 100 would have given us a power easily in excess of 0.8 for a simple linear regression design. The actual experimental effect was an approximate 6% difference between emmetropes and high myopes, but we now calculate that we retain a power of approximately 0.7, despite this reduced effect. When the study was planned this was to be the largest study of its type carried out at that time. In addition, other investigators had used both eyes to report their study numbers but we viewed it more appropriate to use just one eye, given the usual symmetry in size between eyes, thus maintaining the statistical assumption of independent sampling. Finally, other studies had looked at the extreme effects of high myopia and excessive axial length on retinal vessel density, thus assaying pathology and pathological drivers, whereas we sought to determine whether changes across the normal and abnormal spectrum of axial length showed a natural variation/correlation. Thus the inclusion of only ~30% moderate and high myopes in our sample was a deliberate aspect of the experimental design. Whilst we acknowledge the reviewer’s comments, we contend that the experiment remains adequately powered and true to the original experimental question asked.

The authors excluded subjects with retinal pathology but it is also mentioned that myopic degeneration was not excluded in the sampling process. This makes a deviation from the sampling criteria and the authors didn’t mention in which group these participants come under.

We agree with the reviewer that we have not provided details on the number of myopic participants that showed signs of myopic retinal changes (there were 19 subjects displaying myopia-related peripapillary changes across the myopic groups but no other retinal signs were noted). This information is now included in the manuscript (Page 2 of 15, Lines 80-81). We acknowledge that we intentionally excluded known pathology (glaucoma etc.) as these conditions have known effects on microvasculature and anatomy that are likely of a mechanism different to that explored in the current study. The inclusion of subjects with degenerative changes related to myopia reflects the fact that we planned to look at a population in which any vascular changes detected were most likely related to eye size than to a specific pathology. We thank the reviewer for pointing out a potential bias, but would reiterate that the changes were only in the peripapillary region and included as an expected change of the eye size in these individuals and therefore relevant to the study hypothesis.

Age group is not convincing, the authors already mentioned eye elongation extensively occurs during the age of 8-14, then presence of participants up to 50 years again makes a confusion.

We acknowledge the reviewer’s concerns, but would reiterate the highlighted statement in the manuscript (Page 2 of 15, Lines 73-75) that the study was designed to look for deficits in retinal vasculature after maximum period of eye growth had occurred and during a stable period of eye size/refraction and vascular health. Under the simple premise of our experimental hypothesis and design, to assess vascular parameters during expected periods of eye growth would have led to greater complications in interpreting the data and would have necessitated a longitudinal study design. We therefore thank the reviewer for the comments but contend that the study design actually reduces any confusion that might arise in interpreting the data.

In the page 3, Line number 120; it is mentioned that only the data for longer eye of each subject was used; that is not a proper way of analysis

We thank the reviewer for their observation, but argue that it is good practice, and indeed usual, to make an a priori decision on which eye will be analysed in a study of this type. Sometimes that decision will be the left or right eye, and sometimes the least or most myopic eye might be used. We view this as an arbitrary decision, with the main importance that only one eye is used and that the decision does not induce any bias. We can reassure the reviewer that, as would be expected, there was a high level of correlation between pairs of eyes (0.988, 0.989 and 0.969 respectively for axial length, VCD and refractive error) and therefore argue that the outcome of data analysis would have been extremely similar whether right, left, longest or shortest eye were used.

Reviewer 3 Report

This manuscript by Khan et al. reported the finding of the  relationship between the axial eye size and the density of the retinal microvasculature in the macular region. This is a complete and original report with solid experimental evidence by using more advanced OCTA imaging technique.

I only have one minor question, I was wondering whether there is any gender, racial or age-related differences in this study? I understand that the authors may not be sufficiently enough to address this question  due to the sample size in this study.   

I would like the authors to give  a brief discussion regarding whether there is any gender, racial or age-related differences in their retinal microvasculature density study or cite some references that showed the retinal microvasculature may associated with age?  There is some references showed that the retinal microvasculature density related to age, please see the following references.

1.    Seung Hun ParkHeeyoon ChoSun Jin HwangBeomseo JeonMincheol SeongHosuck YeomMin Ho KangHan Woong Lim,* and Yong Un Shin* Changes in the Retinal Microvasculature Measured Using Optical Coherence Tomography Angiography According to Age. J Clin Med. 2020 Mar; 9(3): 883.

2.    Yantao WeiHong JiangYingying ShiDongyi QuGiovanni GregoriFang ZhengTatjana RundekJianhua Wang. Age-Related Alterations in the Retinal Microvasculature, Microcirculation, and Microstructure. Investigative Ophthalmology & Visual Science. 2017, Vol.58, 3804-3817.

Author Response

Reviewer 3

Comments and Suggestions for Authors

This manuscript by Khan et al. reported the finding of the  relationship between the axial eye size and the density of the retinal microvasculature in the macular region. This is a complete and original report with solid experimental evidence by using more advanced OCTA imaging technique.

I only have one minor question, I was wondering whether there is any gender, racial or age-related differences in this study? I understand that the authors may not be sufficiently enough to address this question  due to the sample size in this study.

We thank the reviewer for their suggestion. We have analysed the data on the basis of both age and gender and found no significant relationship between vascular density and either of these parameters (see response to Reviewer 1 above) and have modified the manuscript accordingly to reflect this analysis (Page 6 of 15, Lines 8-10). Whilst we also compared the vascular density for the range of ethnicities identified by subjects in the study, we found no significant differences but feel the analysis is not particularly meaningful, given the very small numbers in a number of the ethnic groups, thus we have not included this analysis in the revised manuscript.

I would like the authors to give  a brief discussion regarding whether there is any gender, racial or age-related differences in their retinal microvasculature density study or cite some references that showed the retinal microvasculature may associated with age?  There is some references showed that the retinal microvasculature density related to age, please see the following references.

We thank the reviewer for the provision of the references and for the suggestion that we discuss our findings in this respect in the manuscript. We have now included the outcomes of our further analysis in the results section (Page 6 of 15, Lines 8-10) and in the discussion (Page 12 of 15, Lines 206 to 210), cited both references and included these in the reference list.

  1. Seung Hun Park, Heeyoon Cho, Sun Jin Hwang, Beomseo Jeon, Mincheol Seong, Hosuck Yeom, Min Ho Kang, Han Woong Lim,* and Yong Un Shin* Changes in the Retinal Microvasculature Measured Using Optical Coherence Tomography Angiography According to Age. J Clin Med. 2020 Mar; 9(3): 883.

2.    Yantao Wei; Hong Jiang; Yingying Shi; Dongyi Qu; Giovanni Gregori; Fang Zheng; Tatjana Rundek; Jianhua Wang. Age-Related Alterations in the Retinal Microvasculature, Microcirculation, and Microstructure. Investigative Ophthalmology & Visual Science. 2017, Vol.58, 3804-3817.

Round 2

Reviewer 2 Report

The comments from the authors are reasonable and I beleive this makes the article suitable for reconsidering for publication.